# Rapid Test for Adulteration of Fritillaria Thunbergii in Fritillaria Cirrhosa by Laser-Induced Breakdown Spectroscopy

**DOI:** 10.3390/foods12081710

**Published:** 2023-04-20

**Authors:** Kai Wei, Geer Teng, Qianqian Wang, Xiangjun Xu, Zhifang Zhao, Haida Liu, Mengyu Bao, Yongyue Zheng, Tianzhong Luo, Bingheng Lu

**Affiliations:** 1School of Optics and Photonics, Beijing Institute of Technology, Beijing 100081, China; 3120170346@bit.edu.cn (K.W.); 3120205322@bit.edu.cn (X.X.); 3120215335@bit.edu.cn (Z.Z.); 3120195328@bit.edu.cn (H.L.); 3120210588@bit.edu.cn (M.B.); 3220210509@bit.edu.cn (Y.Z.); 3220210516@bit.edu.cn (T.L.); 3120220567@bit.edu.cn (B.L.); 2Key Laboratory of Photonic Information Technology, Ministry of Industry and Information Technology, Beijing Institute of Technology, Beijing 100081, China; 3Department of Engineering Science, Institute of Biomedical Engineering, University of Oxford, Oxford OX3 7LD, UK; 4Yangtze Delta Region Academy of Beijing Institute of Technology, Jiaxing 314033, China

**Keywords:** laser-induced breakdown spectroscopy, quantitative analysis, chemometric methods

## Abstract

Fritillaria has a long history in China, and it can be consumed as medicine and food. Owing to the high cost of Fritillaria cirrhosa, traders sometimes mix it with the cheaper Fritillaria thunbergii powder to make profit. Herein, we proposed a laser-induced breakdown spectroscopy (LIBS) technique to test the adulteration present in the sample of Fritillaria cirrhosa powder. Experimental samples with different adulteration levels were prepared, and their LIBS spectra were obtained. Partial least squares regression (PLSR) was adopted as the quantitative analysis model to compare the effects of four data standardization methods, namely, mean centring, normalization by total area, standard normal variable, and normalization by the maximum, on the performance of the PLSR model. Principal component analysis and least absolute shrinkage and selection operator (LASSO) were utilized for feature extraction and feature selection, and the performance of the PLSR model was determined based on its quantitative analysis. Subsequently, the optimal number of features was determined. The residuals were corrected using support vector regression (SVR). The mean absolute error and root mean square error of prediction obtained from the quantitative analysis results of the combined LASSO-PLSR-SVR model for the test set data were 5.0396% and 7.2491%, respectively, and the coefficient of determination R^2^ was 0.9983. The results showed that the LIBS technique can be adopted to test adulteration in the sample of Fritillaria cirrhosa powder and has potential applications in drug quality control.

## 1. Introduction

Fritillaria has high medicinal value because of its ability to moisten the lungs, relieve cough, reduce phlegm, and it can also be consumed as a food [1,2]. Owing to the difference in efficacy and supply and demand, the price of Fritillaria cirrhosa is several times higher than other Fritillaria varieties [3]. In the market, unscrupulous merchants sometimes mix cheaper Fritillaria powders with Fritillaria cirrhosa powder and attempt to sell them off as pure Fritillaria cirrhosa. Therefore, it is necessary to develop a rapid and convenient method to test the level of adulteration of other Fritillaria powders in Fritillaria cirrhosa powder samples [4]. In this paper, the adulteration of Fritillaria thunbergii powder in Fritillaria cirrhosa powder was tested.

Currently, these techniques, such as atomic absorption spectrometry (AAS), inductively-coupled plasma-mass spectrometry (ICP-MS), and inductively-coupled plasma optical emission spectrometry (ICP-OES), can be utilized to quantitatively analyse pharmaceutical products; however, these conventional analysis techniques require complex and time-consuming pre-treatments, such as high temperature and high acid digestion [5]. Fluorescence, Raman, and near-infrared spectroscopies are the most prevalent spectroscopic detection techniques. When adopting these techniques, sample processing is complex and time consuming, and weak spectral signals are susceptible to background light, which prevents in situ, real-time, and rapid online detection [6].

Laser-induced breakdown spectroscopy (LIBS) is an atomic emission spectroscopy technique with the advantages of in situ, real-time, rapid, and simultaneous multi-element detection [7,8]. The LIBS technique is applied in various fields, such as biological tissue detection [9,10,11], explosives detection [12,13,14], coal analysis [15,16,17], polymer identification analysis [18,19,20], food analysis [21,22,23], alloy analysis [24,25,26], ore identification analysis [27,28,29], and soil element detection [30,31,32].

Research on the applications of LIBS techniques in drug testing has also garnered significant attention from scholars globally [33,34]. Peichao Zheng et al. used LIBS technique to quantify Pb and Cu elements in Ligusticum wallichii, a Chinese medicine, and they established multiple linear regression models between the LIBS spectral intensities and the mass fractions of Pb and Cu, respectively; accordingly, the obtained limit of detections for Pb and Cu were 15.7 μg/g and 6.3 μg/g, respectively [35]. Maogen Su et al. studied the solubility of metallic elements during the decoction of Astragalus membranaceus using LIBS technique. The concentration of Cd in the solution was calculated using the calibration-free LIBS (CF-LIBS) method, and the results of the CF-LIBS were verified via the internal standard method. The results obtained from the study showed that the concentrations of Cd calculated by these two methods were within a 10% error margin [36]. Fei Liu et al. adopted the LIBS technique to quantify five nutrient elements in Panax ginseng samples from eight origins. The least squares support vector machine (LS-SVM) model, based on least absolute shrinkage and selection operator (LASSO), was the best predictor for K, Ca, Mg, Zn, and B elements in the samples, and the obtained root mean square error of prediction (RMSEP) values were 0.7704 mg/g, 0.0712 mg/g, 0.1000 mg/g, 0.0012 mg/g, and 0.0008 mg/g, respectively [5]. G. Dastgeer et al. conducted a semi-quantitative study on the elements in calcium tablets using the LIBS technique. The elements, Ca, Mg, Fe, and Zn, were detected in the experimental samples, and their approximate contents were determined from the spectral intensity of each element. Further, the results of the study were consistent with the range of the elements in the instructions [37]. H. Asghar et al. conducted quantitative analysis of sage samples via the CF-LIBS method. Accordingly, elemental spectral lines for Fe, Ca, Ti, Co, Mn, Ni, and Cr were detected in the sage samples. The concentrations of these elements calculated using the CF-LIBS method were 48.1%, 0.7%, 5.3%, 8%, 11%, 12.3%, and 14.6%, respectively [38]. However, to the best of our knowledge, no report has been presented in the literature on the rapid detection of adulterated amounts of pharmaceuticals via the LIBS technique.

Herein, we first test the level of adulteration in Fritillaria cirrhosa powder sample using LIBS techniques. According to the different Fritillaria cirrhosa and Fritillaria thunbergii content, 21 experimental samples with different adulteration levels were prepared, and their LIBS spectra were obtained. Partial least squares regression (PLSR) was adopted as the quantitative analysis model and four data normalization methods, namely, mean centring (MC), normalization by total area (NA), standard normal variable (SNV), and normalization by the maximum (NM), which were used to compare the performance on the PLSR models. Principal component analysis (PCA) and LASSO were utilized to extract and select features from the data, as well as to quantitatively analyse the performance of the PLSR model. Accordingly, the optimal number of features was determined, and the residuals were corrected using SVR. The LASSO method has been applied in the rapid origin identification of chrysanthemum morifolium [39] and detection of exogenous contamination of metal elements in lily bulbs [40] using LIBS. However, it has not been used for quantification combined with PLSR and SVR. We also make the comparison between it and the PCA method, which proves that it can achieve ideal results in Chinese traditional medicine analysis.

## 2. Materials and Methods

### 2.1. Experimental Sample Preparation

The experimental setup employed in this study was the same as that used in a previous study [3]. The adopted *Fritillaria cirrhosa* and *Fritillaria thunbergii* powder samples were purchased from Anhui Tong Huatang Chinese Herbal Beverage Technology Co. (Bozhou, China) and Sichuan Haorui Gallium Biotechnology Co. (Chengdu, China). In the experimental procedure, the powdered *Fritillaria cirrhosa* and *Fritillaria thunbergii* were first mixed into experimental samples with 21 different adulteration levels, as listed in Table 1. Subsequently, for each sample, the powdered samples were continuously and vigorously vibrated at 1000 rpm for 10 h using a multifunctional vortex mixer (VM-500Pro, Joanlab, Huzhou, China). Next, the powdered samples were retrieved, placed on weighing paper, and weighed with an electronic balance (BSA124S-CW, Sartorius, Shanghai, China); accordingly, a 0.5-g weight was obtained. The samples were then pressed for 5 min at a 20-MPa pressure using a tablet press (HY-12 Tianjin, Tianguang Optical Instruments Co., Ltd., Tianjin, China) to obtain disc shaped samples with a diameter and thickness of 13 and 2 mm, respectively, and each sample with different doping levels was pressed into two slices, as illustrated in Figure 1. Finally, the pressed samples were placed on a three-dimensional motorized translation table, and 100 LIBS spectra were obtained at different positions for each sample; accordingly, 200 spectra were obtained for two tablets of each sample, and then two spectra were averaged into one spectrum. Finally, 100 LIBS spectra were obtained for each experimental sample with a different doping level.

### 2.2. LIBS Experiments

The main LIBS setup of our experiments has been shown in the previous work [3]. A homemade Q-switched Nd: YAG laser (1064 nm, 30 mJ, 1 Hz, 10 ns) is used as the excitation source. The laser beam is reflected by three mirrors and then focused onto the sample surface with a NIR-corrected microscopic objective (Mitutoyo, 10X, working distance is 30.5 mm). The plasma radiation is collected and focused into a 600 μm fiber using convex lenses. Then, the LIBS spectra are analyzed by the fiber spectrometer (AvaSpec 2048-2-USB2, Avantes) with a range of 190~1100 nm and a resolution of 0.2~0.3 nm.

## 3. Results and Discussion

### 3.1. LIBS Spectra of the Samples

The typical LIBS spectra of 21 randomly selected experimental samples with different doping levels are presented in Appendix A. The Ca, Na, K, H, O, and N elements, including the CN and C_2_ molecular bands, were identified in the experimental samples and labelled in the LIBS spectra of Sample 1. From Appendix A, the intensities of H, O, N, CN, and C_2_ in the LIBS spectra of the experimental samples with different doping levels are independent of the doping amount of Fritillaria thunbergii, which is primarily influenced by air. To eliminate the interference of airborne components on the intensities of the LIBS spectra of the experimental samples, non-metallic elements were not used in subsequent analyses. The wavelength of 300–800 nm and intensities greater than 1000 counts of Ca (393.3 nm, 396.8 nm, and 422.6 nm), Na (588.9 nm and 589.5 nm), and K (766.4 nm and 769.8 nm) for three metallic elements are presented in Table 2 for the seven LIBS spectral lines.

### 3.2. Quantitative Analysis Modelling

In this study, the data point intensities of seven spectral lines for each spectrum of three metal elements were adopted as input variables (66 variables in total), 100 LIBS spectra from each of Samples 4, 6, 8, 14, 16, and 18 were used as the test set, and 100 LIBS spectra from each of the remaining 15 samples were utilized as the training set to develop the PLSR model.

#### 3.2.1. Data Standardization

Data normalization can reduce the fluctuations between the LIBS spectra of Fritillaria and improve the performance of quantitative analysis models. The effects of the MC, NA, SNV, and NM methods on the performance of the quantitative models were discussed. The mean absolute error (MAE) and RMSEP of the test set obtained using the PLSR model under the four data normalization methods are listed in Table 3.

From Table 3, the MAE values of the test set obtained using the PLSR quantitative analysis model under the four data normalization methods of MC, NA, SNV, and NM are 24.2832%, 48.0711%, 8.6604%, and 8.6111%, respectively; the RMSEP values of the test set obtained are 26.6806%, 60.7073%, 10.8970%, and 10.8760%, respectively. Among the four data normalization methods, the MAE and RMSEP values of the PLSR model’s test set were the lowest by pre-processing the data using the NM method. The relationship between the predicted and standard values of the test set obtained by the PLSR quantitative analysis method under NM data normalization is illustrated in Figure 2; evidently, the MAE and RMSEP values of the test set are 8.6111% and 10.8760%, respectively, and the coefficient of determination, R^2^, of the test set’s fitted straight line is 0.9651. The error bars in Figure 3 represents the standard deviation (SD) of the predicted value based on measurements. We calculated the SD value of Equation (1) in all the figures of this work.
(1)$$S=\sqrt{\frac{1}{n-1}\sum_{i=1}^{n}{\left(x_{i}-\bar{x}\right)}^{2}}$$
where the $\bar{x}$ represent the average value of sample $x_{i}$, *i* is the integer index, and *n* is the number of samples.

#### 3.2.2. Feature Variable Selection

To reduce the amount of data input, discard noise and unimportant features, and to achieve the objective of improving the speed of data processing, it is necessary to select feature variables. Feature variable selection includes two types of feature extraction and feature selection. Additionally, in this study, we adopted PCA and LASSO for feature extraction and feature selection, respectively. LASSO is a compressed estimate. It obtains a more refined model by constructing a penalty function, so that it compresses some coefficients and sets some coefficients to zero. Therefore, Lasso retains the advantages of subset contraction, and it is an estimation of processing with compound linear data.

After normalizing the data via the NM method, the feature variables were extracted from the LIBS spectral data using the PCA method. Figure 3 presents the variance ratio of each principal component (PC) and the cumulative variance ratio obtained after the feature extraction using the PCA method.

As illustrated in Figure 3, the percentage of variance of each individual PC gradually decreased, and the percentage of cumulative variance gradually increased as the number PC increased. The variance ratios of the first three PCs were 89.77%, 7.53%, and 1.17%. The first PC contained most of the original information, and the cumulative variance ratio of the first three PCs reached 98.47%.

To investigate the effect of the number of PCs on the performance of the PLSR quantitative analysis model, the PC scores were adopted as the inputs to the PLSR model for modelling, and the relationship between the RMSEP of the training set and the number of PCs is illustrated in Figure 4.

From Figure 4, the RMSEP of the training set fluctuates with the increase in PC number, exhibiting a general decreasing trend. When the number of PC is 64, the lowest RMSEP value is obtained via the PLSR quantitative analysis model. Therefore, the first 64 PCs are chosen for the test set data analysis. The relationship between the predicted and standard values of the test set obtained by using the PLSR quantitative analysis method for the first 64 PCs as input data is illustrated in Figure 5. From Figure 5, the MAE and RMSEP values of the test set are 7.1585% and 9.1709%, respectively, and the coefficient of determination, R^2^, of the test set’s fitted straight line is 0.9920.

To investigate the effect on the performance of the PLSR model after adopting LASSO feature selection for quantitative analysis, the LIBS spectral data of the samples were first standardized using the NM method; subsequently, the importance of the data after standardizing the LIBS spectra was evaluated using the LASSO method, and the importance weight values of 66 variables were obtained. Finally, the importance weight values of the 66 variables were normalized, and the obtained normalized weight values based on the training set data are presented in Figure 6.

Figure 6 shows the normalized weight values of 66 variables obtained for the importance assessment of the LIBS spectral lines using the LASSO method. Among the 66 spectral features, nine of them have normalized weight values of 0, thus indicating that these nine features do not play a role in the quantitative analysis. Meanwhile, the elements corresponding to the top 57 wavelengths in importance ranking, and their importance weights are listed in Table 4.

To investigate the effect of the number of features on the performance of the PLSR quantitative analysis model, the 57 spectral features with non-zero normalized weight values were ranked according to the normalized weight values from the largest to the smallest. The different numbers of spectral features, corresponding to the top 1–57 normalized weight values, were adopted as inputs to the PLSR model, respectively, and the relationship between the RMSEP of training set and the number of spectral features is illustrated in Figure 7.

As illustrated in Figure 7, the RMSEP of the training set fluctuates with the increase in the number of spectral features, exhibiting a general decreasing trend. The RMSEP value obtained using the PLSR quantitative analysis model is lowest when the number of input spectral features is 57. Then, these 57 features are used in the test data analysis. The relationship between the predicted and standard values of the test set using the PLSR quantitative analysis model for the first 57 spectral features as input data is illustrated in Figure 8. From Figure 8, it can be seen that the MAE and RMSEP values of the test set are 7.1038% and 9.1523%, respectively, and the coefficient of determination, R^2^, of the test set’s fitted straight line is 0.9728.

Compared with the PCA-PLSR model, the MAE and RMSEP of the predicted values of the LASSO-PLSR model’s test set were relatively better, decreasing from 7.1585% to 7.1038% and from 9.1709% to 9.1523%, respectively.

#### 3.2.3. Residual Correction

The residual values were obtained by subtracting the predicted values of the LASSO-PLSR model from the standard values and were corrected using SVR. The SVR is a common kind of traditional machine learning method, which has extraordinary performance in the small sample training. In SVR, the straight line required for fitting data becomes a hyperplane. The SVR creates an “interval band” on both sides of the linear function, which does not calculate the loss for all samples falling into the interval zone. The samples outside the interval zone are recognized as support vectors. That is, only the support vector will affect its function model. Finally, the optimized model is obtained by minimizing total loss and maximum interval.

The relationship between the mean value of the residuals of the 100 spectra of each modelled sample and the mean value of the intensity of the most important spectral line (Na I 589.6 nm) is illustrated in Figure 9.

As illustrated in Figure 9, the data points comprising the average of the residuals of 100 spectra of each modelled sample and the average of the intensity of the most important spectral line (Na I 589.6 nm) are randomly distributed and exhibit a nonlinear relationship overall. To further improve the accuracy of the quantitative analysis model, the most prevalent nonlinear kernel function in SVR (radial basis kernel function) was adopted to develop a SVR-based residual correction model with optimal parameters (*c* = 76.7312, *g* = 9.3966), and the predicted value of the PLSR model plus the predicted value of the SVR correction model was considered as the final predicted value. The relationship between the predicted and standard values of the test set, based on the LASSO-PLSR-SVR residual correction model, is illustrated in Figure 10; from this figure, it can be seen that the MAE and RMSEP values of the test set are 5.0396% and 7.2491%, respectively, and the coefficient of determination, R^2^, of the test set’s fitted straight line is 0.9983.

From Figure 8 and Figure 10, it can be seen that the performance of the LASSO-PLSR-SVR residual correction-based model improved better than the LASSO-PLSR-based model in the quantitative analysis of the Fritillaria LIBS spectral data. The MAE value of the test set decreased from 7.1038% to 5.0396%, the RMSEP value decreased from 9.1523% to 7.2491%, and the coefficient of determination R^2^ of the test set’s fitted straight line increased from 0.9728 to 0.9983. The obtained results indicate that the residual correction model can compensate for the limitations of the conventional PLSR model in elucidating nonlinear characteristic variables and ultimately improve the prediction accuracy of the quantitative analysis model.

## 4. Conclusions

In this study, we proposed the adoption of the LIBS technique, combined with machine learning methods in the rapid test of adulteration levels in Fritillaria cirrhosa powder. Using PLSR as a quantitative analysis model, the effects of four data normalization methods, MC, NA, SNV, and NM, on the quantitative analysis performance of the PLSR model, were compared, among which the NM data normalization method exhibited the best performance. Accordingly, the effects of feature extraction and feature selection on the quantitative analysis performance of the PLSR model using PCA and LASSO were compared and analysed, and the performance of the LASSO-PLSR-based model was to be relatively better. A correction model for the residuals was further developed using SVR to compensate for the limitations of the conventional PLSR model in elucidating the nonlinear feature variables. The final LASSO-PLSR-SVR-based residual correction model improved the accuracy of the quantitative analysis relative to the LASSO-PLSR model, and the MAE of the test set decreased from 7.1038% to 5.0396%, the RMSEP decreased from 9.1523% to 7.2491%, and the coefficient of determination R^2^ of the test set’s fitted straight line increased from 0.9782 to 0.9983. The experimental results demonstrated the effectiveness of adopting the LIBS technique combined with machine learning in the rapid test of adulteration levels in Fritillaria cirrhosa powder. Furthermore, this study has important application value for drug testing, regulation, and quality control.

## Figures and Tables

**Figure 1 foods-12-01710-f001:**
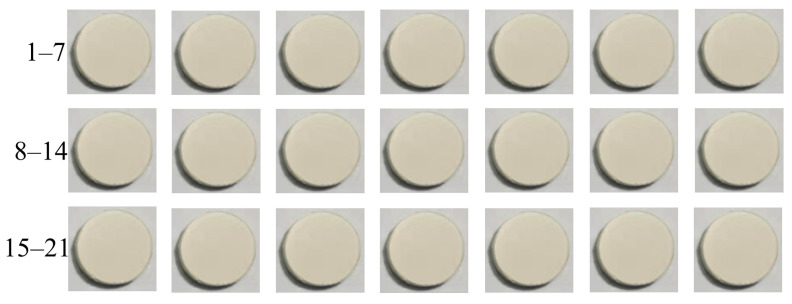
Physical diagram of the experimental samples with 21 different doping levels.

**Figure 2 foods-12-01710-f002:**
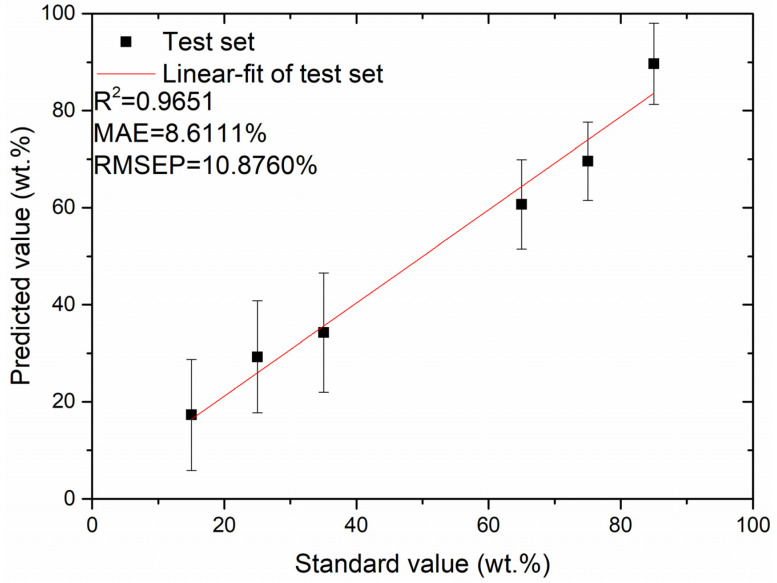
Plot of relationship between predicted and standard values of the test set obtained using the PLSR quantitative analysis method under NM data normalization.

**Figure 3 foods-12-01710-f003:**
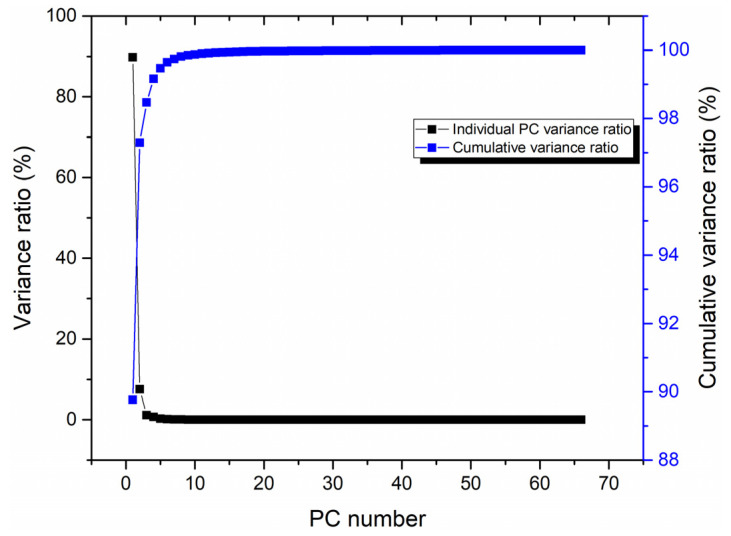
Variance ratio of each PC and cumulative variance ratio.

**Figure 4 foods-12-01710-f004:**
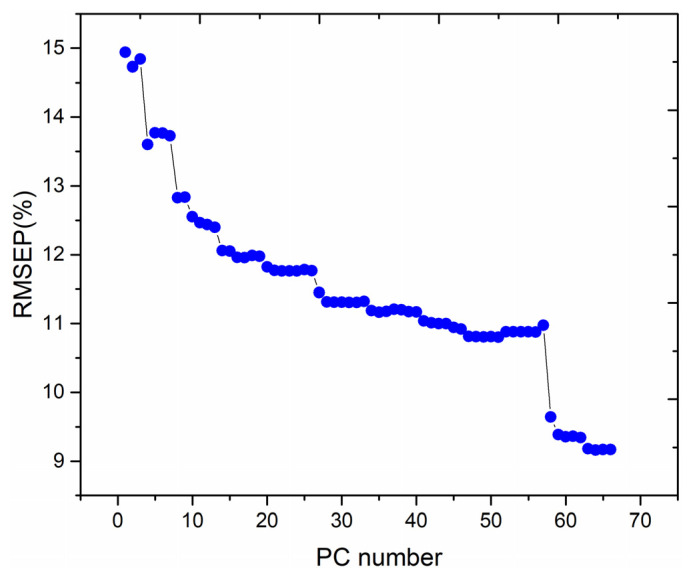
Effect of the number of PCs on the performance of the PLSR quantitative analysis model.

**Figure 5 foods-12-01710-f005:**
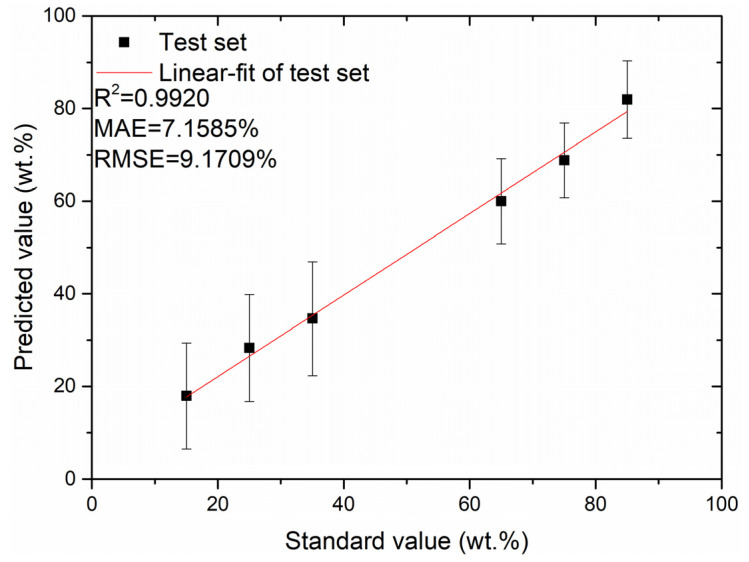
Plot of the relationship between the predicted and standard values of the test set obtained for the first 64 PCs as inputs to the PLSR model.

**Figure 6 foods-12-01710-f006:**
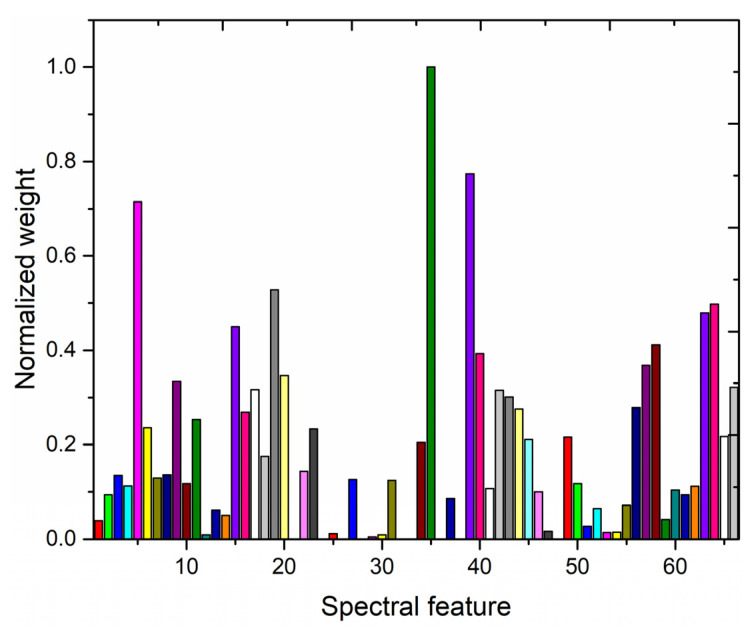
Normalized weight values of 66 variables obtained for the importance assessment of the LIBS spectral lines using the LASSO method.

**Figure 7 foods-12-01710-f007:**
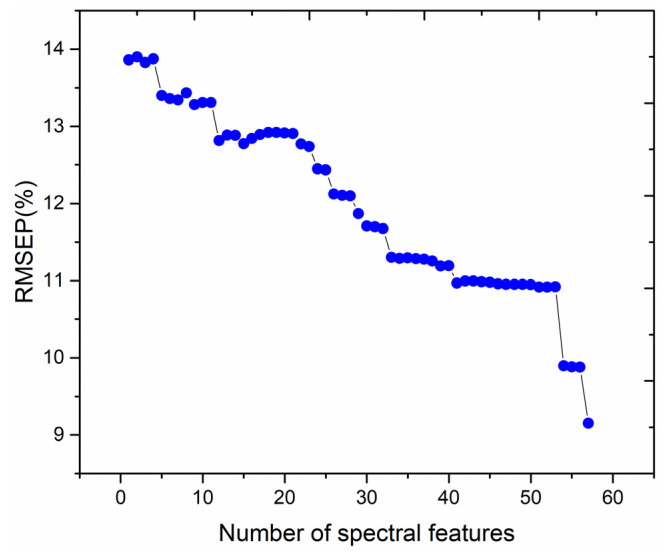
Effect of the number of spectral features on the performance of the PLSR quantitative analysis model.

**Figure 8 foods-12-01710-f008:**
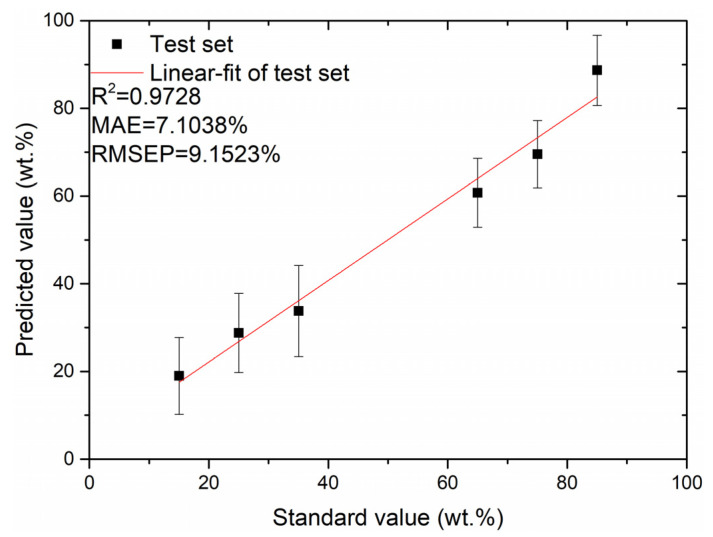
Plot between the predicted and standard values of the test set for the first 57 spectral features as inputs to the PLSR model.

**Figure 9 foods-12-01710-f009:**
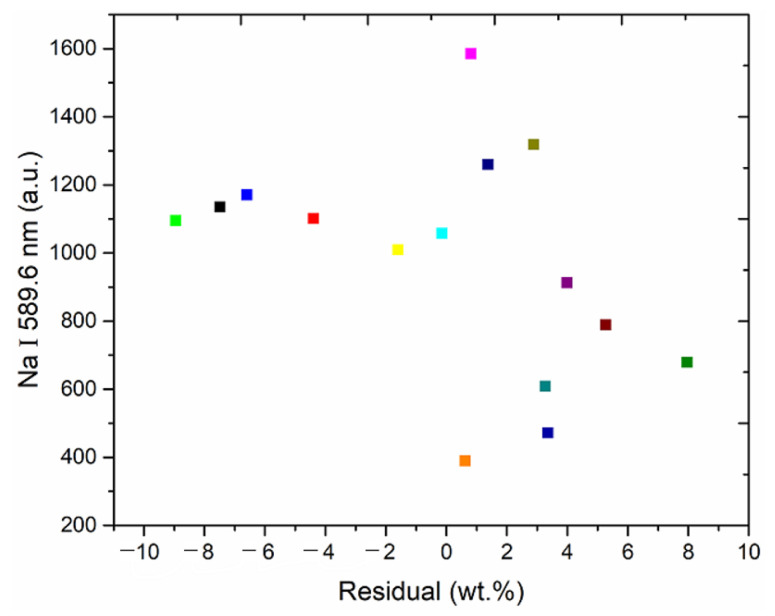
Plot of the average of the residuals of 100 spectra of each modelled sample against the average of the intensities of the most important spectral lines (Na I 589.6 nm).

**Figure 10 foods-12-01710-f010:**
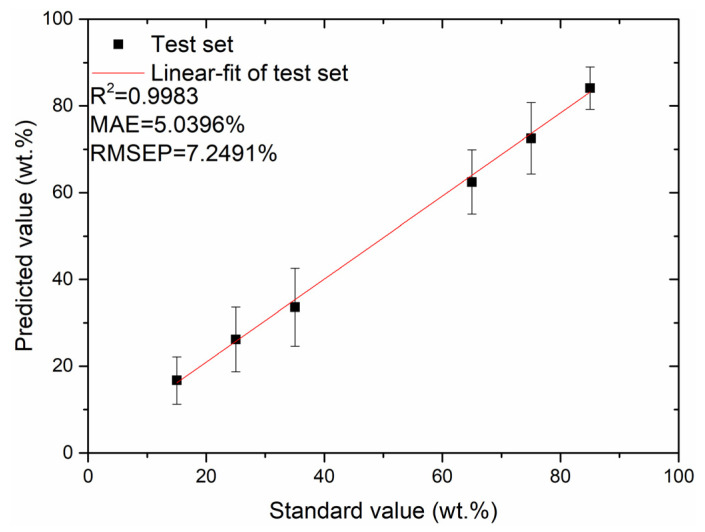
Plot of relationship between the predicted and standard values of the test set based on the LASSO-PLSR-SVR residual correction model.

**Table 1 foods-12-01710-t001:** Experimental samples of 21 different doping levels.

Sample Number	*Fritillaria cirrhosa* (g)	*Fritillaria thunbergii* (g)	*Fritillaria thunbergii* Content (%)
1	0.0000	1.0000	100.0000
2	0.0502	0.9501	95.9351
3	0.1001	0.9002	89.9810
4	0.1505	0.8505	85.0020
5	0.2008	0.8005	79.9621
6	0.2500	0.7500	75.0150
7	0.3008	0.7001	70.0030
8	0.3507	0.6501	65.0105
9	0.4002	0.6005	59.9560
10	0.4505	0.5506	55.0105
11	0.5003	0.5005	49.9900
12	0.5506	0.4503	45.0005
13	0.6001	0.4008	39.9920
14	0.6503	0.3500	35.0420
15	0.7001	0.3000	30.0530
16	0.7506	0.2500	25.0000
17	0.8009	0.2007	20.0539
18	0.8507	0.1501	15.0350
19	0.9008	0.1003	10.0070
20	0.9503	0.0507	5.0185
21	1.0040	0.0000	0.0000

**Table 2 foods-12-01710-t002:** Three metal elements and their wavelengths in the experimental samples.

Element	Ca II	Ca II	Ca I	Na I	Na I	K I	K I
Wavelength (nm)	393.3	396.8	422.6	588.9	589.5	766.4	769.8

**Table 3 foods-12-01710-t003:** MAE and RMSEP of the test set obtained using the PLSR model under the four data normalization methods.

Data Normalization Methods	MC	NA	SNV	NM
MAE (%)	24.2832	48.0711	8.6604	8.6111
RMSEP (%)	26.6806	60.7073	10.8970	10.8760

**Table 4 foods-12-01710-t004:** Elements and importance weights corresponding to the top 57 wavelengths in importance ranking.

Order of Importance	Wave Length (nm)	Element	Importance Weights	Order of Importance	Wave Length (nm)	Element	Importance Weights
1	589.6	Na I	1.0000	30	393.0	Ca II	0.1348
2	590.2	Na I	0.7738	31	396.4	Ca II	0.1290
3	393.6	Ca II	0.7149	32	588.6	Na I	0.1261
4	587.2	Na I	0.5281	33	589.1	Na I	0.1243
5	770.7	K I	0.4977	34	767.0	K I	0.1172
6	770.4	K I	0.4790	35	421.5	Ca I	0.1171
7	422.9	Ca I	0.4503	36	393.3	Ca II	0.1123
8	769.1	K I	0.4119	37	770.2	K I	0.1116
9	764.4	K I	0.3929	38	764.7	K I	0.1067
10	768.9	K I	0.3683	39	769.7	K I	0.1038
11	587.5	Na I	0.3464	40	766.0	K I	0.1001
12	397.0	Ca II	0.3338	41	769.9	K I	0.0939
13	771.2	K I	0.3215	42	392.7	Ca II	0.0935
14	423.5	Ca I	0.3163	43	589.9	Na I	0.0858
15	764.9	K I	0.3150	44	768.3	K I	0.0714
16	765.2	K I	0.3010	45	767.6	K I	0.0645
17	768.6	K I	0.2787	46	422.4	Ca I	0.0613
18	765.5	K I	0.2757	47	422.6	Ca I	0.0502
19	423.2	Ca I	0.2688	48	769.4	K I	0.0416
20	421.8	Ca I	0.2527	49	392.5	Ca II	0.0391
21	396.1	Ca II	0.2355	50	767.3	K I	0.0272
22	588.0	Na I	0.2331	51	766.2	K I	0.0163
23	771.0	K I	0.2170	52	768.1	K I	0.0149
24	766.8	K I	0.2160	53	767.8	K I	0.0142
25	765.7	K I	0.2109	54	588.3	Na I	0.0116
26	589.5	Na I	0.2045	55	588.9	Na I	0.0091
27	423.8	Ca I	0.1751	56	422.1	Ca I	0.0090
28	587.8	Na I	0.1438	57	588.8	Na I	0.0046
29	396.7	Ca II	0.1358				

## Data Availability

Data is contained within the article or Appendix A.

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
