# Peer review of "Rapid Test for Adulteration of Fritillaria Thunbergii in Fritillaria Cirrhosa by Laser-Induced Breakdown Spectroscopy"

_foods, 2023, doi:10.3390/foods12081710_

Round 1

Reviewer 1 Report

The manuscript describes the development of a rapid test for detecting adulteration in Fritillaria using the LIBS technique and chemometric analyses with spectral normalization and feature selection to improve quantification performace. Twenty-one samples with different levels of adulteration were prepared and divided into training (15) and test (6) sets. One thing that caught my attention and, if confirmed, would be a flaw in the analysis, is that the authors applied the test data to select the best parameters for quantification. This can be seen in the data from the graphs in figures 6, 8, 9, and 11. In this case, the final result obtained may be influenced by overfitting. The correct approach would be to select such parameters using the training data and only perform external validation testing at the end to check the technique's robustness.

I would like the authors to clarify how parameter selection was carried out in this work - was it based on the test or training data set? It is not clear from the text when the training data were used. Regardless, I recommend either a reanalysis of the data if the authors used test data for parameter selection or an improvement in the text writing if the data were correctly analyzed.

Some minor corrections are listed below:

- Describe the characteristics of the LIBS equipment used.

- Table 1 has an error in the last column (the percentages are reversed).

- I don't think it's necessary to show the 21 LIBS spectra separately (Figure 2). They can be combined into one graph or put in the supplementary material.

- Figure 3: how were the error bars calculated?

- Briefly explain how the LASSO and SVR methods work.

Reviewer 2 Report

The paper deals with statistical analysis of LIBS spectra collected in order to detect adulteration of a vegetable powder used in traditional medicine and food by addition of the same ingredient from a cheeper plant of the same family (Fritillaria).

The paper is clear and well written, a bit boring in the didactic description of the figure and table in the text, which for instance makes completely useless table 3. But this is a matter of style and can stay as presented.

The only serious concern that I have is about bibliography. There is a very large overwiev of LIBS analytical applications to quantitative analysis of any kind of samples, some based also on combined application of different types of statistical analysis. While in the text there is a novelty statement about the use of LIBS to detect medicine adulteration, there is not any mention about the former use of the same PCA and LASSO, either alone or combined, in the reported references. It is also not straightforward to understand is the application of SVM for LASSO residual correction is a novelty (absolute novelty, or at least novelty for drug analysis) or a routine procedure, in the latter case the specific reference should be quoted.

I am not suggesting to add further references. Probably a better quoting, where relevant, of what already reported would be sufficient to improve the presentation of the paper, so stressing novelty aspects with respect to consolidated procedure (including in the latter the normalization on maxima which is widely used).

In my opinion the paper can be minor changes, after better quoting references relevant to the statistical data analysis presented.

Round 2

Reviewer 1 Report

my doubts were clarified.